# Multidisciplinary and Coordinated Management of Osteoporotic Vertebral Compression Fractures: Current State of the Art

**DOI:** 10.3390/jcm13040930

**Published:** 2024-02-06

**Authors:** Khalid Al Taha, Nicolas Lauper, David E. Bauer, Andreas Tsoupras, Enrico Tessitore, Emmanuel Biver, Dennis E. Dominguez

**Affiliations:** 1Spine Team, Division of Orthopaedics and Trauma Surgery, Geneva University Hospitals, Faculty of Medicine, University of Geneva, 1211 Geneva, Switzerland; khalid.al-taha@hcuge.ch (K.A.T.); nicolas.lauper@hcuge.ch (N.L.); dudi.bauer@gmail.com (D.E.B.); andreas.tsoupras@hcuge.ch (A.T.); 2Division of Neurosurgery, Geneva University Hospitals, Faculty of Medicine, University of Geneva, 1211 Geneva, Switzerland; enrico.tessitore@hcuge.ch; 3Division of Bone Diseases, Geneva University Hospitals, Faculty of Medicine, University of Geneva, 1211 Geneva, Switzerland; emmanuel.biver@hcuge.ch

**Keywords:** osteoporotic vertebral compression fracture, low back pain, osteoporotic fracture classification, thoracolumbar fracture in older adults, percutaneous vertebral augmentation

## Abstract

Osteoporotic vertebral compression fractures (OVCFs) present a significant health concern, affecting a substantial portion of the older adult population worldwide. This narrative review explores the prevalence, diagnostic challenges and management strategies for OVCFs. Despite the increasing incidence and impact on morbidity and mortality, existing clinical guidelines lack consistency and clear diagnostic and therapeutic recommendations. The review addresses key questions faced by physicians dealing with older adult patients experiencing acute back pain, offering insights into triage, radiological assessments and classification systems. We propose a comprehensive algorithm for clearing OVCF, considering clinical presentation, radiological findings and morphological aspects. Emphasis is placed on the importance of medically treating osteoporosis alongside OVCF management. The review encompasses relevant literature from 1993 to 2023, provides a detailed discussion on triage issues and incorporates a clinically oriented classification system developed by the German Society for Orthopaedics and Trauma. The Material and Methods section outlines the extensive literature search carried out in PUBMED, encompassing clinical and experimental studies, systematic reviews and meta-analyses. The articles retained focused mainly on answering critical questions regarding radiological assessments, imaging modalities and the presence of a specific classification system for OVCFs. The review emphasises that the evaluation and management of OVCFs necessitates a multidisciplinary approach involving spine specialists and bone disease experts. It also addresses the role of conservative versus surgical treatments, with a focus on percutaneous vertebral augmentation. The conclusion summarises the algorithm derived for use in emergency departments and general practice, aiming to streamline OVCF management, reduce unnecessary examinations and ensure optimal patient care. The algorithm recommends primary diagnosis using computed tomography, with magnetic resonance imaging reserved for specific cases. The review advocates a holistic approach, integrating medical and surgical interventions to address the complex challenges posed by OVCFs in ageing populations.

## 1. Background

Osteoporotic vertebral compression fractures (OVCFs) are common among older adult populations with osteoporosis, occurring in approximately 30% to 50% of people over the age of 50 and 12% in women and men aged from 50 to 79 years in the world, affecting an estimated 1.4 million patients annually, and incidence rates rise exponentially with age, especially in women [1,2,3]. Worldwide, approximately 20% of older adults aged more than 70 and 16% of post-menopausal women experience OVCFs [4]. The incidence of OVCFs is 10.7/1000 person years in women and 5.7/1000 person years in men [5]. However, two-thirds to three-quarters of vertebral fractures are not recognised at the time of clinical occurrence [1].

As the older adult population grows, OVCFs are becoming a major source of pain and dysfunction [5] and a significant cause of increased morbidity and mortality in developed countries [6,7]. However, existing clinical guidelines do not contain consistent diagnostic and therapeutic recommendations for managing OVCFs [8]. The present study aimed to answer the most relevant questions for physicians facing patients with acute back pain and dysfunction so that they can clear OVCFs in their daily practice.

This narrative review examines relevant articles published between 1993 and 2023 and discusses them in detail while answering fundamental triage questions encountered in daily clinical practice. We designed an algorithm for clearing osteoporotic spine fractures based on clinical presentation, radiological type and morphological aspects (an osteoporotic fracture classification). We also aimed to highlight the importance of medically treating osteoporosis regardless of the OVCF’s treatment.

Because 5% of adults over 50 years old presenting with acute low back pain (aLBP) and a red flag indicator (>50 years old, anticoagulant use, fever, being immunocompromised, intravenous drug abuse, recent surgery or epidural injection, neurological deficit and trauma) have an OVCF, those red flags should be ruled out [9,10].

After low-energy falls, older adults with potential spine fractures should first undergo a radiological assessment using computed tomography (CT) and then plain radiography [11]. A CT scan has greater sensitivity and specificity than plain radiography, preventing any OVCF injuries from being missed and limiting unnecessary radiological examinations [12]. A negative CT scan is sufficient for clearing the majority of low-back pain (LBP) associated with low-energy fall injuries [12]. In cases involving motor symptoms or radio-clinical discrepancies, the advice of a specialised spine surgeon must be requested. 

Magnetic resonance imaging (MRI) should not be used routinely, and its indication should be discussed with a spine surgeon. Non-traumatic aLBP among older adults is associated with OVCF in 5% [10]. OVCFs are severe osteoporotic fractures since they are highly associated with low bone mineral density (BMD) and are a significant risk factor for subsequent fracture. The risk of secondary fracture after an initial vertebral fracture is about 25% within 2 years and 40% within 5 years [13]. Thus, any patient with a vertebral fracture should be considered in the imminent risk or high-risk groups for new vertebral and non-vertebral fractures, and they require evaluation and treatment [14,15]. Fracture liaison service programmes may help to identify and engage secondary fracture prevention in patients with index vertebral fractures. 

Neglected OVCFs can become a major source of pain and dysfunction and are becoming a significant cause of morbidity and mortality in developed countries [5,6,7]. A better understanding of how the burden of LBP is related to the progression of vertebral collapse in OVCFs is essential. They are often successfully treated conservatively, but the failure of conservative treatment can lead to serious complications, particularly if other risk factors are not dealt with [16].

Optimal patient management requires predetermined protocols for all actions taken in emergency departments (EDs) or in general practice, but existing clinical guidelines give inconsistent diagnostic and therapeutic recommendations for managing osteoporotic fractures [8]. This narrative review will help to simplify screening for treating physicians. It outlines the most relevant literature on clearing OVCFs in older adult patients and suggests an easy-to-use algorithm to minimise the risk of missing OVCFs and initiate adequate medical treatment and surgical management if indicated.

## 2. Material and Methods

An exploratory literature search was performed in PUBMED for English-language articles published from January 1993 to May 2023. Keywords used were “osteoporotic vertebral compression fracture”, “low back pain”, “OF classification”, “thoracolumbar fracture in older adults” and “percutaneous vertebral augmentation”. Clinical and experimental studies, systematic reviews and meta-analyses were all included. Off-topic articles, case reports, letters to the editor, editorials, general commentaries and publications concerning paediatric populations were excluded. The references in the articles reviewed were also checked for relevant studies not yielded by the initial search; they were added if appropriate.

We selected articles containing relevant information to answer the three most common questions when a patient with aLBP and a suspected OVCF is encountered in an ED or in general practice. Does the patient need a radiological assessment? What imaging needs to be requested? If an OVCF is diagnosed, should the next step be conservative or surgical management? After answering these questions, we describe our management algorithm and illustrate cases of OVCF that were treated surgically.

### 2.1. Who Needs Radiological Assessment?

When patients over 50 years old present with a new spontaneous LBP pain or following a low-energy fall, lumbar spine imaging is mandatory. Being over 50 is a red flag in itself and could be associated with a serious pathology like an OVCF. Other red flags suggesting that a radiological assessment is indicated include anticoagulant use, fever, genitourinary issues such as urine retention or sexual dysfunction, being immunocompromised, intravenous drug abuse, recent surgery and epidural injection [9]. Enthoven et al. found that 33% of OVCFs were associated with red flags that were diagnosed as the causes of back pain among older adults presenting in general practice [10].

### 2.2. What Imaging Needs to Be Requested?

In EDs, the first-line diagnostic tool for thoracolumbar fractures in older adult patients with low-energy trauma or new onset spontaneous vertebral pain is usually a conventional biplanar X-ray of the spine. Several previous studies have shown an X-ray’s moderate or poor accuracy, with negative predictive values ranging from 14% to 81% and a negative likelihood ratio of 0.43, indicating that a negative X-ray cannot safely rule out fractures of the thoracic or lumbar spine. Plain radiography’s sensitivity for fracture detection is estimated at 49.2% in the thoracic spine and 57.8% in the lumbar spine, with its specificity varying between 97% and 99%. In comparison, CT has an estimated sensitivity from 95% to 98% and a specificity from 99% to 100% [11,12,17].

Plain radiography’s lack of accuracy is unacceptable in clinical practice, and a multiplanar reconstruction (MPR) CT scan of the thoracolumbar spine must be considered the first-line assessment for such patients [12,18]. A complementary standing biplanar X-ray, centred in the area of interest, should be performed to rule out any initial kyphotic deformity, as these are not always evident on a CT scan in the supine position. This relevant information will influence treatment management. Furthermore, radiological follow-up is performed by comparing successive standing X-rays to document the stability or progression of the vertebral collapse and then deciding on medical or surgical management [12]. 

In cases of neurological compromise, multiple vertebral fractures, suspicion of posterior band injury or the inability to determine the age of the fracture and deformity, MRI scanning is the most suitable diagnostic tool. Fresh OVCFs present with low-intensity areas of the vertebra on T1-weighted sequences and high-intensity areas of the vertebra on T2-weighted sequences [14,15]. MRI helps to detect spinal canal stenosis with dural sac compression, bony oedema, or ligamentous oedema. Oedema, revealed as hyperintensity in fat saturation sequences (STIR, DIXON), is considered a sign of an acute or non-consolidated chronic fracture [12]. MRI also has the advantage of detecting fractures without a vertebral deformity. However, MRI scans can have significant false-positive findings, and, in our opinion, they should not be routinely used for the diagnosis of OVCFs. In cases involving a discrepancy between radiological and clinical findings, MRI can be requested after consulting a spine specialist.

When diagnosing and treating osteoporotic fractures, clinicians should always keep in mind that adult populations with a high prevalence of OVCFs could also have a high prevalence of oncological diseases such as multiple myeloma or metastatic spine tumours. The differential diagnosis of an OVCF and a metastatic fracture is generally difficult using X-ray and CT scanning, and, in such cases, an MRI scan, with and without gadolinium enhancement, is indicated to visualise the posterior border of the vertebral body, extension into the posterior elements and diffuse abnormal bone marrow [19].

Finally, vertebral morphometry is a complementary tool used in some centres for objectively grading vertebral fractures and is already used to calculate predictive indexes for them. Vertebral morphometry could also be useful for selecting non-fractured vertebrae at a high risk of future vertebral fractures [20,21].

### 2.3. Is There a Specific Classification System for Osteoporotic Fractures?

Numerous classification systems for osteoporotic fractures have been proposed in the past, some of which received broad acceptance in the literature [22,23,24]. More recently, the Osteoporotic Fractures working group of the German Society for Orthopaedics and Trauma’s (DGOU) Spine Section developed and proposed a clinically oriented classification system for osteoporotic thoracolumbar fractures [25]. Their osteoporotic fracture (OF) classification system consists of five subgroups classifying fractures according to the amount of collapse, the involvement of the endplates and posterior wall, and the involvement of the posterior tension band (Figure 1). 

Based on their proposed classification, the DGOU also described a scoring system incorporating the initial fracture type and various injury and patient-related factors. The OF score’s primary objective is optimising the decision-making process for selecting the most suitable treatment approach, whether surgical or non-surgical. Surgical treatment is proposed for fractures scoring >6 points, with non-surgical treatment proposed for fractures scoring <6 points [26].

As the authors recommended, the fracture should be re-evaluated as early as one week after the initial presentation to detect any progression of the fracture if treated non-surgically. The timely identification of fractures at a high risk of progression and collapse is of paramount importance to ensure optimal treatment and mitigate potential complications, including subsequent vertebral fractures in this fragile patient group [27]. In this context, Scheyerer et al. conducted a systematic review evaluating clinical and radiographic indicators to predict treatment failures and complications in the conservative management of OVCFs [28]. Their article described patient-specific, fracture-specific, radiological and other factors associated with OVCFs and the adverse outcomes of non-surgical treatment. The studies included in their systematic review had identified various factors associated with different outcomes after non-surgical therapy for osteoporotic fractures.

When considering patient-specific factors, an age > 73 years, a bone mineral density t-score < −2.95 SD, a BMI < 23 kg/m^2^ and a modified frailty index > 2.5 have all been identified as risk factors for conservative treatment failure [29,30].

The presence of certain radiological risk factors plays a significant role in predicting the failure of conservative therapy for OVCFs (Table 1). These risk factors include the appearance of an intervertebral cleft detected in functional or prone-position lateral radiographs, as well as the involvement of both endplates and the posterior wall on conventional radiographs, which increases the likelihood of collapse during conservative treatment. Lastly, evidence of risk factors is related to sagittal spinopelvic malalignment.

Considering other risk factors, a history of bisphosphonate therapy was linked to the appearance of the vertebral cleft sign; however, this did not contribute to reducing pain or improving the quality of life [41].

### 2.4. Medical Management of Pain and Osteoporosis

Clinical history and physical examination are crucial in the initial evaluation of patients with vertebral fractures. The goals are to exclude non-osteoporotic causes of bone fragility, look for other risk factors for bone fragility and detect potentially secondary and remediable causes of osteoporosis. Non-traumatic vertebral fractures might indeed be due to causes other than osteoporosis, which should always be excluded, especially in fractures occurring in non-older adult patients or in cases of vertebral fractures in vertebrae higher than T4. The thoracic disk herniation associated with Scheuermann’s kyphosis should also be differentiated from compression fractures. Secondary causes of osteoporosis (such as hypogonadism, hyperthyroidism, hyperparathyroidism, hypercortisolism, celiac disease and other forms of malabsorption, and mastocytosis) and other causes of low bone mass (such as osteomalacia or myeloma) should also be evaluated and addressed before initiating therapy. Laboratory evaluations are required and will help in this context (Table 2).

Even though an OVCF is an indication for osteoporotic treatment independently of BMD, dual-energy X-ray absorptiometry (DXA) should be performed to measure BMD at the spine (excluding fractured vertebrae) and hip. This will help to refine the risk of future fractures and to monitor the response to anti-osteoporotic drugs. BMD thresholds may also be included in insurance reimbursement criteria for these drugs. 

The pharmacological management of aLBP associated with an OVCF includes acetaminophen, opioids, non-steroidal anti-inflammatory drugs and calcitonin. However, the risk of side effects from these drugs, especially in older adult patients, should be considered, and associated non-pharmacological management is required most of the time (and described below).

The management of osteoporosis is crucial to reducing the risk of subsequent fractures. Despite the imminent risk of a new fragility fracture after an index vertebral fracture, a substantial proportion of patients with osteoporotic fractures—who are eligible for a pharmacological treatment of osteoporosis—remains untreated (a treatment gap of about 70% in Europe in 2019) [42]. Lifestyle factors that contribute to bone loss, including smoking, excessive alcohol, physical inactivity and poor nutrition, should all be addressed. Vitamin D ± calcium supplements (according to dietary intake/intestinal absorption) should be provided. Several anti-osteoporotic drugs are effective, compared with placebo, to reduce the risk of vertebral fractures (39% to 77% effective, depending on treatment) [43,44]. They increase BMD either by inhibiting bone resorption or/and by promoting bone formation (Table 3). In clinical trials, all patients received calcium and vitamin D supplements, and the patient subgroups that achieved the greatest absolute risk reduction for future fractures were those with a prevalent vertebral fracture. Therefore, patients diagnosed with OVCFs should be offered appropriate anti-osteoporotic therapy as soon as possible. Anabolic agents have greater anti-fracture efficacy and produce larger increases in bone density than anti-resorptive drugs [45]. This was shown markedly in the VERO study comparing teriparatide with risedronate and in the ARCH study comparing romosozumab with alendronate [46,47]. Therefore, in the absence of contraindication, most guidelines recommend considering an anabolic drug as the first-line treatment for patients with a very high or imminent risk of fracture, which includes patients with an acute vertebral fracture [48,49,50]. The effects of anabolic agents are transient, so sequential treatment with anti-resorptive drugs following anabolic ones is required. In addition, a meta-analysis of randomised controlled trials associated teriparatide treatment with a reduced risk of new or worsening back pain compared to placebo, hormone replacement therapy or alendronate [51].

### 2.5. Conservative Therapy or Surgical Management for OVCFs?

The presence of an OVCF might seem to suggest bed rest. A comparative study on the treatment of OVCFs using either early ambulation or at least 1 week of absolute bed rest, conducted by a Korean research group, showed higher complication rates in the bed rest group in terms of constipation, indigestion, dizziness and neurasthenia. Therefore, patients with OVCFs treated conservatively should not stay in bed and should walk as soon as possible [52]. Structural changes in bone architecture, mineralisation and muscle atrophy were observed at higher rates in the bed rest group, also leading to a higher risk of non-union [53,54].

In approximately one in five cases, the conservative management of aLBP related to an OVCF can result in such failures as a new OVCF at another level, chronic or persistent LBP pain, progressive kyphotic deformity and/or neurological compromise [55]. Risk factors related to this bad prognosis are greater age, a lower T-score, previous OVCFs, acquired segmental deformity [55] and the radiological risk factors described above.

Another reason for the failure of conservative management is sagittal imbalance, with a distance superior to 65 mm drawn from the centre of the vertebral body to the sagittal vertical axis [38,56]. Poor clinical outcomes are commonly related to pseudoarthrosis, a segmental deformity angle superior to 15°, vertebral comminution or a posterior ligamentous injury; therefore, a close radiological follow-up is mandatory for the early recognition of a failure of conservative management [57].

Bracing could be a complement to conservative management, but a systematic review by Holfer and Jones (2020) concluded that there was no evidence to suggest that a rigid brace was superior to a soft brace or no brace. Furthermore, bracing could also lead to significant physical deconditioning, bedsores and poor therapeutic compliance by the patient [58].

Is there any place for physiotherapy for patients with OVCFs? Physical exercise may have some benefits in terms of pain control, physical function and quality of life outcomes, but no definitive conclusions could be made regarding the benefits of exercise for patients with OVCFs due to the inconsistency and low quality of the evidence [59].

Several reasons were involved in the conservative treatment’s failure to adequately deal with the aLBP related to OVCFs, and in these cases, surgical management should be discussed. The surgical treatment of OVCFs involves percutaneous vertebral augmentation [Figure 2] or vertebroplasty (cement injection only) or correction with minimally invasive spondylodesis [Figure 3], depending on the radiological OF classification and the radiological risk factors mentioned above. The UK’s National Institute for Health and Care Excellence recommends vertebral augmentation procedures and the percutaneous insertion of craniocaudal expandable implants as two treatment options for patients with severe and disabling pain after a recent OVCF [60].

There are also risk factors of percutaneous vertebral augmentation treatment failure in cases of sagittal imbalance with a sagittal vertical axis > 5 cm, a low T-score and fracture mobility [61]. Cement leakage and kyphosis after a primary operation are also risk factors correlated with a subsequent OVCF after percutaneous vertebral augmentation treatment. However, there is no correlation with cement amount, sex, age, BMI or cement injection approaches [62].

## 3. Discussion

OVCFs are the most common insufficiency fractures occurring in older adult populations, and they influence their daily activities and quality of life. Symptoms can vary from being asymptomatic to aLBP occurring spontaneously or subsequent to a low-energy trauma. Acute back pain that increases when changing posture, decreases in the supine position and limits mobility is suggestive of an OVCF [63]. Pain is typically localised in the midline spine but can also be felt in a unilateral or bilateral pattern into the flank, anterior abdomen or posterior superior iliac spine [64].

A diagnosis can be made using history, the presence of any red flags (age > 50 years old, anticoagulant use, fever, being immunocompromised, intravenous drug abuse, recent surgery or epidural injection, neurological deficit and trauma [9]), physical examination (presence of any tenderness over vertebrae or neurological assessment) and imaging [65]. When a family or ED physician has a history and physical examination compatible with a possible OVCF, a radiological assessment should be performed. Plain X-ray is often prescribed spontaneously, but nearly 50% of OVCFs may be missed [11,12,17]. For this reason, this narrative review aims to increase awareness of the initial imaging requirements for patients presenting with spontaneous aLBP, with or without a low-energy fall, and/or red flags, i.e., a multiplanar reconstruction (MPR) CT scan or at least a standing X-ray. Indeed, Pedersen et al. conducted a meta-analysis to determine plain radiography’s accuracy in detecting fractures among older adults after low-energy falls. Sensitivity was only between 40.0% and 72.3% [11]. VandenBerg et al. conducted another meta-analysis of blunt thoracolumbar-spine trauma evaluation in the ED and the diagnostic accuracy of patient history, physical examination and imaging. They found a pooled sensitivity from 56% to 64% and a pooled specificity from 97% to 99% with plain radiography, and a pooled sensitivity from 95% to 98% and a pooled specificity from 99% to 100% with CT [17]. A study published in the JAMA in 2020, examining proportions of imaging workflows, showed that only 23% of patients with thoracic trauma and 35% of patients underwent a CT scan as their first imaging [18].

This narrative review aims to encourage a change in the initial imaging for patients presenting with spontaneous aLBP or after a low-energy fall and red flags. The diagnostic accuracy of plain radiographs has been proven insufficient, missing between 27.7% and 60% of clinically significant fractures and, thus, requiring a CT scan. Multidetector or multiplanar reconstruction (MPR) CT scans show strong evidence of being the safest way to rule out a thoracolumbar injury with high sensitivity and specificity [17].

The utility of MRI for clearing thoracolumbar injury after screening using CT is still uncertain, mainly due to a high rate of false positives. As consensus has yet to be found in the literature, its utility mandates further studies. STIR/DIXON sequences may be useful for screening posterior tension band injuries or bone oedema or in patients with multiple-level fractures and those with a suspicion of a pathological vertebral fracture [12,19]. We conclude that the routine use of MRI for clearing is not beneficial. In very rare cases with discrepancies between imaging and clinical findings indicating a pathological neurological status, an MRI could be requested on the advice of a spine specialist.

After radiological assessment with evidence of an OVCF, a close collaboration between spine specialists and bone disease specialists is mandatory. Laboratory assessments (Table 3) should be performed to target the medical treatment. On the spine surgery side, osteoporotic thoracolumbar fractures’ radiological aspects are classified according to the OF classification system developed by the Osteoporotic Fractures working group of the German Society for Orthopaedics and Trauma’s (DGOU) Spine Section [18]. This OF classification consists of five groups. In association with the OF classification, there is an OF score that helps the spine surgeon to determine whether management should be by conservative or surgical treatments (OF scores of >6 or <6, respectively) [19]. Physicians should, nevertheless, also consider the risk factors for failure in conservatively treated OVCFs, as described in the systematic review by Scheyrer et al. [22]. The OVCF’s characteristics influence whether the treatment should be conservative or surgical, and if it is surgical, which procedure should be performed.

There is a high rate of failure in osteoporotic vertebral fractures managed conservatively. Petitt et al. mentioned a failure rate of 17.5%, defined by rescue surgery, kyphosis progression, chronic pain and diminished functional condition. Of these, 36.7% had a new fracture, 32% experienced chronic pain and 31.4% showed kyphosis progression and neural structure compression [55].

Vertebral augmentation is the most common treatment for OVCFs, but evidence of its efficacy has been controversial in prior randomised clinical trials (INVEST and Australian Trial) in terms of long-term pain relief, functional outcomes and increased secondary fracture risk [62,66,67]. However, Beall et al. reported levels I and II evidence of kyphoplasty’s superiority over conservative treatment for OVCFs in terms of pain [68], and Hoyt et al. determined that surgical management led to a significant reduction in mortality among patients with OVCFs secondary to physical deconditioning [69]. Hinde et al. confirmed these findings in a meta-analysis of more than 2 million OVCF cases, with patients who underwent surgical management by vertebral augmentation being 22% less likely to die within 10 years than cases managed non-surgically [70]. It is now accepted that vertebral augmentation is a safe surgical procedure of a short duration, offering long-term pain relief, early ambulation, shorter hospital length of stay, lower analgesic use, improved quality of life and greater patient satisfaction [71].

Several types of implants can be used for vertebral augmentation for OVCFs, such as cement alone, a diamond stent with or without cement, an oval stent with or without cement, and a coil or a cube or a spine jack with cement. A 20-year review of experimental biomechanical studies on these different types of implants (Figure 4) used for percutaneous vertebral augmentation showed that the spine jack was best in terms of vertebral body height restoration and maintenance [72]. We recommend correction of the segmental kyphotic deformity and vertebral body height restoration, which, if not corrected, could lead to chronic compensatory LBP.

After considering every aspect of the management of OVCFs, we were able to develop an algorithm for clearing these fractures in cases of spontaneous aLBP or after a low-energy trauma. The algorithm aims to prevent missed OVCFs and to make physicians aware of the need for multidisciplinary collaboration. Avoiding unnecessary complementary examinations reduces costs and improves care management in EDs. Indeed, OVCFs, similar to low-energy trauma hip fractures, are a multifactorial problem engendering a variety of consequences on patients’ general health, and multidisciplinary approaches are important for obtaining the best clinical, functional and psychological results for our patients [73,74]. Multidisciplinary osteoporosis care management is associated with earlier treatment initiation and appreciably lower rates of repeat fracture and mortality [74,75].

## 4. Conclusions

Based on the current literature, we designed and developed simplified and full Osteoporotic Vertebral Compression Fracture (OVCF) management algorithms (Figure 5 and Figure 6) for use in the emergency department or general practice. We aimed to derive an algorithm that would leave no OVCF unidentified while simplifying patient management and avoiding unnecessary evaluations. Computed tomography (CT) scanning should be the first examination prescribed due to its superior sensitivity and specificity. A negative CT scan is sufficient for clearing OVCF except when it highlights problems with gross motor skills, at which point a magnetic resonance imaging scan must be performed. In our opinion, the clinical assessment of patients over 50 years old and the presence of red flags remain pivotal in any evaluation. In such cases, and in cases with positive CT scans, the advice of a spinal surgeon must be obtained to determine the ensuing care.

## Figures and Tables

**Figure 1 jcm-13-00930-f001:**
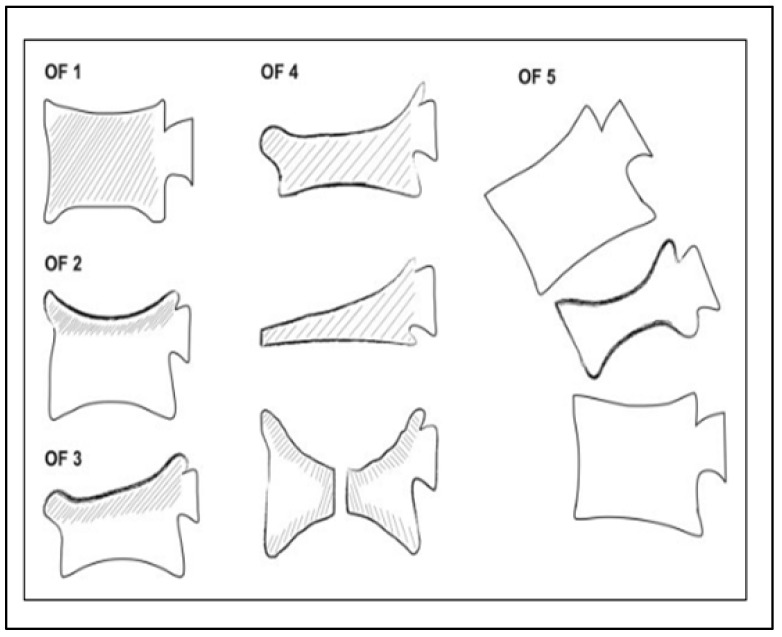
Schematic representation of the five OF classification system subtypes [25].

**Figure 2 jcm-13-00930-f002:**
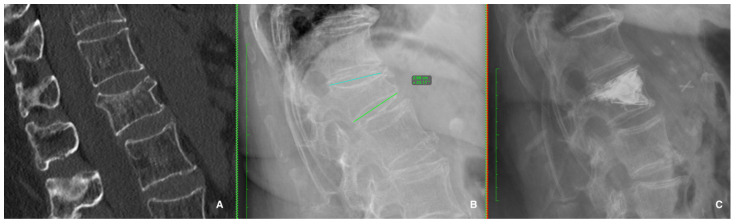
CT scan image (**A**) and standing X-ray image (**B**) show the patient’s 20° kyphotic deformity. Postsurgical X-ray (**C**) shows a stand-alone percutaneous vertebral reconstruction with a spine jack and cement injection in the L1 type 3 OF.

**Figure 3 jcm-13-00930-f003:**
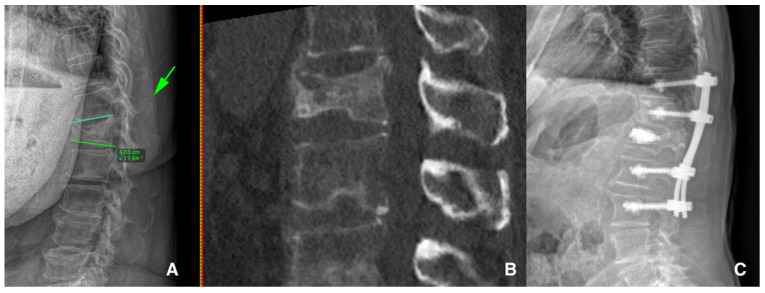
Standing X-ray (**A**) shows the patient’s 17° kyphotic deformity of the vertebral body with a spinous process fracture (green arrow) type 5 OF. CT scan (**B**) shows the upper and lower endplate components. Postoperative standing X-ray (**C**) shows the percutaneous vertebral reconstruction and augmented posterior fixation.

**Figure 4 jcm-13-00930-f004:**
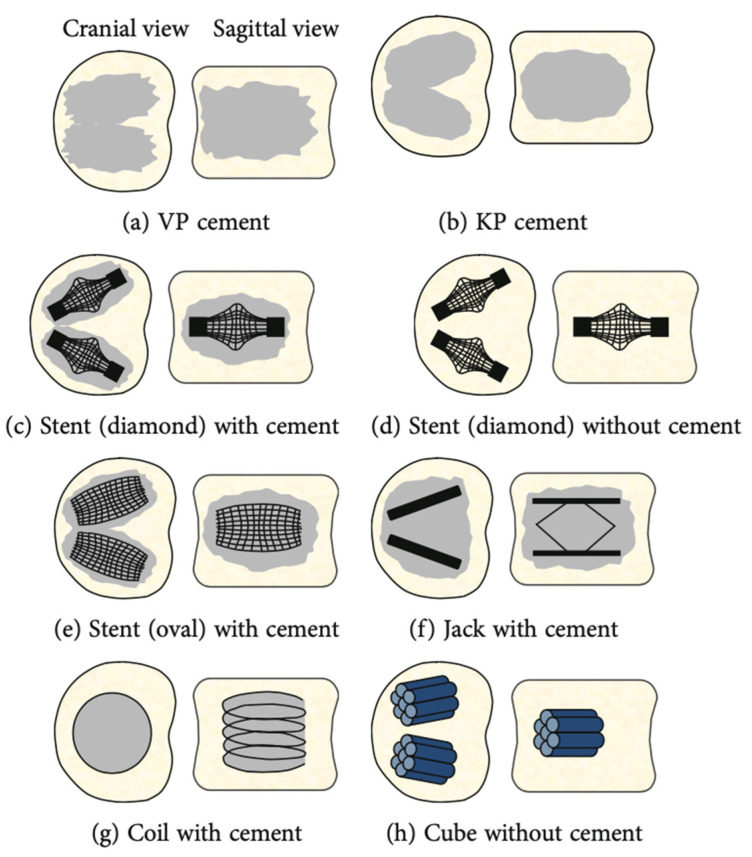
Illustration of the types of implants used in vertebral augmentation [72].

**Figure 5 jcm-13-00930-f005:**
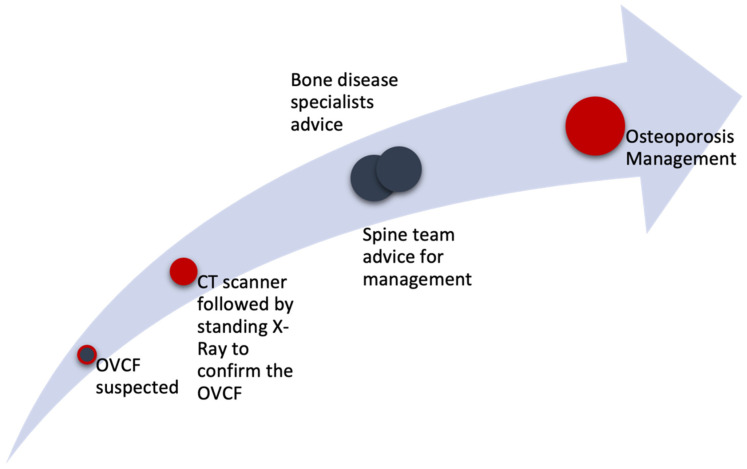
The simplified Osteoporotic Vertebral Compression Fracture management algorithm in our hospital.

**Figure 6 jcm-13-00930-f006:**
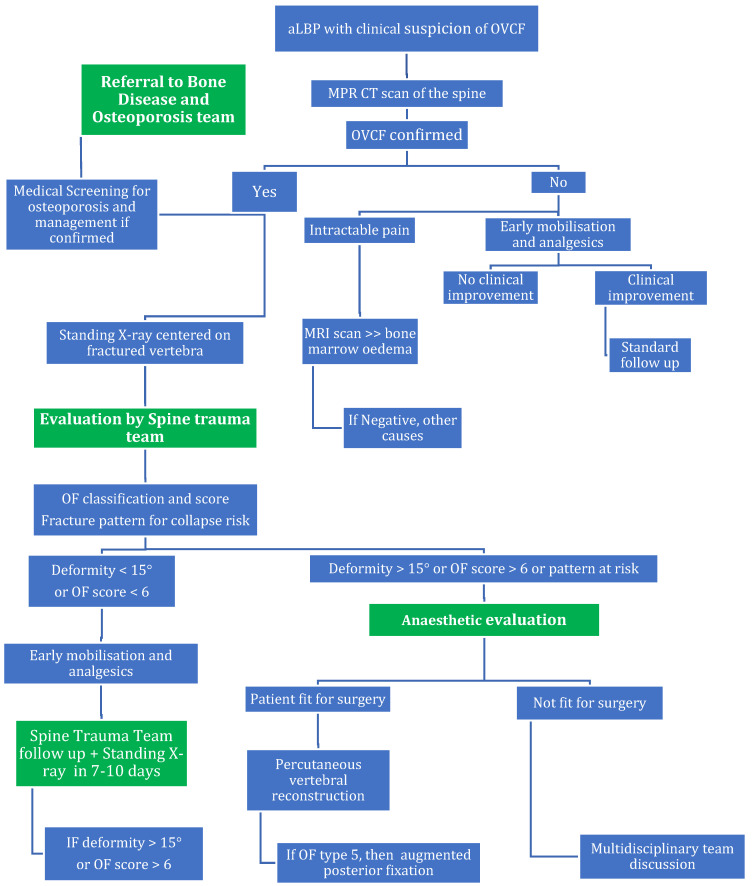
The full Osteoporotic Vertebral Compression Fracture management algorithm for aLBP patients. Abbreviations: aLBP = acute low back pain; OVCF = osteoporotic vertebral compression fracture; MRI = magnetic resonance imaging.

**Table 1 jcm-13-00930-t001:** Risk factors leading to conservative treatment failure.

Study	Related Complication	Risk Factors
Park H.Y. et al. (2018) [31] Hayashi T. et al. (2016) [32]	Vertebral collapse and spinal canal compromise and/or delayed neurological deficit	Posterior wall involvement, initial height loss, midportion-type fractures, swelled-front-type fractures, complex fracture morphology, intravertebral cleft development, fracture instability, fractures at the thoracolumbar junction and specific MRI changes
Sugita M. et al. (2005) [23] Hoshino M. et al. (2013) [33]	Poor clinical outcome	Bow-shaped type fractures, projecting type fractures, swelled-front-type fractures and posterior wall fractures
Goldstein S. et al. (2016) [34] Ha K.Y. et al. (2013) [35] Patil S. et al. (2014) [36]	Progressive vertebral collapse or progressive kyphotic deformity	Location at thoracolumbar junction, midportion-type fractures, intervertebral cleft development, specific fracture types, posterior wall fractures, superior endplate involvement, anterior wall fractures and previous adjacent level fractures
Nakamae T. et al. (2017) [37]	Symptomatic intravertebral cleft	Posterior wall fracture and vertebral instability
Ohnishi T. et al. (2018) [38] Iwata A. et al. (2017) [39] Smorgick Y. et al. (2020) [40]	Poor outcome (pain) and subsequent fracture	Values outside of specific ranges associated - C7 plumb line to the centre of the fractured vertebral body (dynamic sagittal vertical axis) > 6 cm and PT > 27° - PI-LL mismatch > 30° - Sagittal vertebral axis over 50 mm PI lower or higher than 40° to 60°)
Zhang JN, He X, Fan Y et al. (2019) [29] Lee H.M., Park S.Y., Lee S.H. et al. (2012) [30]	Poor outcome (pain control)	age > 73 years, bone mineral density with a t-score < −2.95 SD, BMI < 23 kg/m^2^ and a modified frailty index > 2.5

**Table 2 jcm-13-00930-t002:** Proposed laboratory evaluations for patients with vertebral fractures.

Basic Laboratory Tests for All Patients with Vertebral Fracture	Additional Laboratory Tests according to the Patient’s Clinical History, Physical Examination and Basic Laboratory Tests
- calcium, phosphorous, albumin - total protein - creatinine - alanine and aspartate aminotransferase, gamma-glutamyl transferase, alkaline phosphatase - 25-hydroxyvitamin D - complete blood count - serum protein electrophoresis	- 24 h urine for calcium and creatinine - 24 h urinary free cortisol - anti-transglutaminase antibody - parathyroid hormone (PTH) - tryptase - bone turnover markers: beta-CrossLaps (CTX), type 1 procollagen N-terminal propeptide (P1NP) *

* May help to monitor response and compliance to therapy, but increased after a fracture.

**Table 3 jcm-13-00930-t003:** Pharmacological agents for the treatment of post-menopausal osteoporosis.

Drug and Bone Mechanism of Action	Main Side Effects and/or Contraindications
Inhibitors of bone resorption: (a) Bisphosphonates (BPs) - Alendronate 70 mg/week (oral) - Risedronate 35 mg/week (oral) - Ibandronate 150 mg/month (oral) or 3 mg/3 months (IV) - Zoledronate 5 mg/year (IV) (b) Anti-RANKL monoclonal antibodies - Denosumab 60 mg/6 months (SC) (c) SERMs - Raloxifene 60 mg/day (oral) - Bazedoxifene 20 mg/day (oral)	- Oral BPs: oesophageal and gastric irritation - IV BPs: flu-like symptoms - BPs: chronic renal failure (contraindicated or not recommended if eGFR < 30 mL/min) - Denosumab: dermatological side effects and skin infection - BPs/Denosumab: hypocalcaemia - BPs/Denosumab: osteonecrosis of the jaw (high doses) - BPs/Denosumab: atypical femoral fractures (long-term use) - SERM: deep venous thrombosis
Activators of bone formation: (d) Parathyroid hormone - Teriparatide 20 µg/day (SC) - PTH(1-84) 100 µg/day (SC) - Abaloparatide 80 µg/day (SC)	- Contraindicated in conditions with increased bone turnover (pre-existing hypercalcaemia, primary hyperparathyroidism, Paget’s disease of the bone, unexplained elevation of alkaline phosphatase) - Contraindicated after radiotherapy to bone - Not recommended in monoclonal gammopathy of undetermined significance
Mixed effect: Activators of bone formation and Inhibitors of bone resorption (e) Anti-sclerostin monoclonal antibodies: - Romosozumab 210 mg/month (SC)	- Contraindicated in patients with a history of myocardial infarction or stroke

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
