# Peer review of "Multidisciplinary and Coordinated Management of Osteoporotic Vertebral Compression Fractures: Current State of the Art"

_jcm, 2024, doi:10.3390/jcm13040930_

Round 1
Reviewer 1 Report
Comments and Suggestions for Authors
Dear Editors and authors,
I am grateful to revise this interesting article that I found very useful for clinical practice.
The case of osteoporotic vertebral fractures is still a challenging issue that remains difficult to be faced at different level. Thus, I think that increasing current literature about the management of fractures could be crucial.
Anyway, in my opinion, the Authors should revise their manuscript taking into account the following suggestions in order to render it more suitable for publication in this journal.
General comments: English form need to be revised. In several cases, the syntax of the manuscript is not correct. In particular, the authors should revise punctuation
· In my opinion, the Authors should report and discuss the potential role of vertebral morphometry in the assessment of the vertebral fracture. (Diacinti D, Guglielmi G. Vertebral morphometry. Radiol Clin North Am. 2010 May;48(3):561-75. doi: 10.1016/j.rcl.2010.02.018. PMID: 20609892; Diacinti D, Pisani D, Del Fiacco R, Francucci CM, Fiore CE, Frediani B, Barone A, Bartalena T, Cattaruzza MS, Guglielmi G, Diacinti D, Romagnoli E, Minisola S. Vertebral morphometry by X-ray absorptiometry: which reference data for vertebral heights? Bone. 2011 Sep;49(3):526-36. doi: 10.1016/j.bone.2011.05.027. Epub 2011 Jun 6. PMID: 21672644). In fact, although both CT scan and MRI remain fundamental to study vertebrae, vertebral bone morphometry could help clinicians to obtain important information.
· As for the discussion, in order to improve the quality of their manuscript, the Authors should highlight in a more extensive manner the importance of a multidisciplinary approach to promote the correct treatment of osteoporotic vertebral fracture as well as the prevention of other imminent fractures , citing all the following available studies. Capozzi A, Scambia G, Pedicelli A, Evangelista M, Sorge R, Lello S. Clinical management of osteoporotic vertebral fracture treated with percutaneous vertebroplasty. Clin Cases Miner Bone Metab. 2017 May-Aug;14(2):161-166. doi: 10.11138/ccmbm/2017.14.1.161. Epub 2017 Oct 25. PMID: 29263726; PMCID: PMC5726202;
Skorupski N, Alexander IM. Multidisciplinary osteoporosis management of post low-energy trauma hip-fracture patients. J Am Assoc Nurse Pract. 2013 Jan;25(1):3-10. doi: 10.1111/1745-7599.12002. PMID: 23279273;
Osuna PM, Ruppe MD, Tabatabai LS. FRACTURE LIAISON SERVICES: MULTIDISCIPLINARY APPROACHES TO SECONDARY FRACTURE PREVENTION. Endocr Pract. 2017 Feb;23(2):199-206. doi: 10.4158/EP161433.RA. Epub 2016 Nov 16. PMID: 27849385.
Best regards
Comments on the Quality of English Language
Dear Editors and authors,
I am grateful to revise this interesting article that I found very useful for clinical practice.
The case of osteoporotic vertebral fractures is still a challenging issue that remains difficult to be faced at different level. Thus, I think that increasing current literature about the management of fractures could be crucial.
Anyway, in my opinion, the Authors should revise their manuscript taking into account the following suggestions in order to render it more suitable for publication in this journal.
General comments: English form need to be revised. In several cases, the syntax of the manuscript is not correct. In particular, the authors should revise punctuation
· In my opinion, the Authors should report and discuss the potential role of vertebral morphometry in the assessment of the vertebral fracture. (Diacinti D, Guglielmi G. Vertebral morphometry. Radiol Clin North Am. 2010 May;48(3):561-75. doi: 10.1016/j.rcl.2010.02.018. PMID: 20609892; Diacinti D, Pisani D, Del Fiacco R, Francucci CM, Fiore CE, Frediani B, Barone A, Bartalena T, Cattaruzza MS, Guglielmi G, Diacinti D, Romagnoli E, Minisola S. Vertebral morphometry by X-ray absorptiometry: which reference data for vertebral heights? Bone. 2011 Sep;49(3):526-36. doi: 10.1016/j.bone.2011.05.027. Epub 2011 Jun 6. PMID: 21672644). In fact, although both CT scan and MRI remain fundamental to study vertebrae, vertebral bone morphometry could help clinicians to obtain important information.
· As for the discussion, in order to improve the quality of their manuscript, the Authors should highlight in a more extensive manner the importance of a multidisciplinary approach to promote the correct treatment of osteoporotic vertebral fracture as well as the prevention of other imminent fractures , citing all the following available studies. Capozzi A, Scambia G, Pedicelli A, Evangelista M, Sorge R, Lello S. Clinical management of osteoporotic vertebral fracture treated with percutaneous vertebroplasty. Clin Cases Miner Bone Metab. 2017 May-Aug;14(2):161-166. doi: 10.11138/ccmbm/2017.14.1.161. Epub 2017 Oct 25. PMID: 29263726; PMCID: PMC5726202;
Skorupski N, Alexander IM. Multidisciplinary osteoporosis management of post low-energy trauma hip-fracture patients. J Am Assoc Nurse Pract. 2013 Jan;25(1):3-10. doi: 10.1111/1745-7599.12002. PMID: 23279273;
Osuna PM, Ruppe MD, Tabatabai LS. FRACTURE LIAISON SERVICES: MULTIDISCIPLINARY APPROACHES TO SECONDARY FRACTURE PREVENTION. Endocr Pract. 2017 Feb;23(2):199-206. doi: 10.4158/EP161433.RA. Epub 2016 Nov 16. PMID: 27849385.
Best regards
Author Response
Dear colleague,
Thanks for the revision and suggestions.
I have put in the manuscript and discussed the role of vertebral morphometry and the importance of multidisciplinary management. I have added the articles suggested too and made a global correction of the english by an english speaker and expert in scientific articles.
Best regards.
Dr Khalid Al Taha
Reviewer 2 Report
Comments and Suggestions for Authors
This is a well written narrative review developed with much effort by the authors. Below are my minor comments:
- please adjust the tables fotn to match the text
- authors should specify how many people did the article reviews and their selection
- biases should be acknowledged in the discussion section
- authors should provide a PRISMA flowchart of the methodology they used
- references are adequate (they require a reformating according to our journal)
- english is decent
Author Response
Dear colleague,
Thanks for the revision and suggestions.
I have made a global correction of the english by an english speaker and expert in scientific articles.
The footnotes of all tables and the discussion were revised
The references were adjusted to what is requested by the journal
Articles selection and review is described in detail in the material and method section.
Thanks again for your revision and best regards.
Kindly,
Dr Khalid Al Taha